# Handling Long-Term Safety and Uncertainty in Safe Reinforcement Learning

**Jonas Günster**[1]       **Puze Liu**[2]       **Jan Peters**[1,2]       **Davide Tateo**[1]

[1] TU Darmstadt, Germany
[2] German Research Center for AI, Germany

**Abstract:** Safety is one of the key issues preventing the deployment of reinforcement learning techniques in real-world robots. While most approaches in the Safe Reinforcement Learning area do not require prior knowledge of constraints and robot kinematics and rely solely on data, it is often difficult to deploy them in complex real-world settings. Instead, model-based approaches that incorporate prior knowledge of the constraints and dynamics into the learning framework have proven capable of deploying the learning algorithm directly on the real robot. Unfortunately, while an approximated model of the robot dynamics is often available, the safety constraints are task-specific and hard to obtain: they may be too complicated to encode analytically, too expensive to compute, or it may be difficult to envision a priori the long-term safety requirements. In this paper, we bridge this gap by extending the safe exploration method, ATACOM, with learnable constraints, with a particular focus on ensuring long-term safety and handling of uncertainty. Our approach is competitive or superior to state-of-the-art methods in final performance while maintaining safer behavior during training. [1]

**Keywords:** safe reinforcement learning, chance constraints, distributional RL

## 1  Introduction

Safety is one of the most important problems when applying Reinforcement Learning (RL) to real-world applications, as it is fundamental to prevent RL agents from harming people or causing damage. To deal with this issue, RL researchers developed Safe Reinforcement Learning (SafeRL) techniques, to learn policies maximizing the task performance while satisfying the safety requirements. Safe Exploration (SafeExp) aims to ensure the agent's safety during the exploration phase and formulates the safety problem as a stepwise constraint that the agent should not violate.

Solving the SafeExp problem requires additional prior knowledge, such as constraints, robot dynamics, or previously collected datasets. While the SafeExp algorithms have been deployed successfully in complex real-world tasks [1, 2, 3, 4], most of these approaches suffer from many drawbacks preventing their application to complex or out-of-the-lab tasks. First, safety specifications defined as constraints entail an in-depth understanding of the environment and dynamics. Designing and validating safety constraints requires extensive expertise and experience. Second, real-world applications contain various uncertainty sources, such as sensor noise, model error, environmental disturbance, and partial observability, which are often neglected in the design of constraints. Third, the optimization techniques for constrained optimization problems are limited and often require a specific problem structure, such as quadratic programming with linear complementarity constraints. Lastly, the learning algorithms should also guarantee *Long-Term Safety*, which not only ensures the safety of the current step but also considers the safety of future trajectories. The term *Long-Term Safety*

---

[1]Code is publicly available at `https://github.com/cube1324/d-atacom`

8th Conference on Robot Learning (CoRL 2024), Munich, Germany.

unifies the Forward Invariance property for the static environment with known dynamics [5, 6] and the predictive safety in a dynamic environment without full knowledge of dynamics.

We argue that incorporating prior knowledge for safety-critical robotics applications can be beneficial, as prior knowledge, such as kinematics and dynamics, is often well-studied and readily available. Furthermore, in robotics, we can assume that a sufficiently good approximation of the dynamics model is available, while the complete model of the environment is not. Under this assumption, we combine techniques from model-free SafeRL methods and SafeExp approaches, showing how to exploit the knowledge of the dynamics while learning of unknown, long-term constraints. To achieve this goal, we extend the Acting on the TAngent Space of the COnstraint Manifold (ATA-COM) approach by dropping key assumptions that the constraints are predefined, allowing learning of long-term safety constraints directly from data. Furthermore, we explicitly model the constraints' uncertainty in a distributional RL perspective, which provides us with a way to estimate the total uncertainty of the model. Consequently, we introduce Distributional ATACOM (D-ATACOM), allowing us to derive a risk-aware policy by restricting the level of accepted risk.

Our experiments demonstrate that D-ATACOM achieves a safer performance during the training phase and reaches a similar or better performance at the end of training. For tasks where the optimal policy stays within constraints, D-ATACOM explores more cautiously at the cost of slower learning speed. Instead, D-ATACOM show faster and safer behaviors during training whenever there is a conflict between the policy optimization problem and the constraint satisfaction one.

**Related Work**    In the last decades, SafeRL is a field of increasing interest for deploying learned safe agents to the real world. Constrained Markov Decision Processes (CMDP) [7, 8] framework as a first attempt from RL researchers has gained significant progress in recent research in solving constrained control problems. One important formulation of constraint is the expected cumulative cost. The RL agent aims to maximize the expected return while maintaining the expected cost below a threshold [9, 10, 11, 12, 13, 14, 15, 16, 17]. This type of constraint has been extended to different variants, such as the risk-sensitive constraint [18, 19, 20, 21] and the probabilistic constraint [22, 23, 24]. Different types of constrained optimization techniques are applied in the policy update process, such as the trust-region method [9, 20], the interior point method [12], and the Lagrangian relaxation method [7, 11, 13, 14, 18, 19]. Furthermore, the Lyapunov function is also used to derive a policy improvement procedure [10, 25, 26]. Notably, learning the value function of the cumulative cost has gained incremental attraction as it addresses long-term safety. Recent works have shared a common view that the value function of constraint provides a predictive estimation of safety, such as the feasibility value [27, 28], control barrier function [29, 30], and safety critic [31, 21]. In this paper, we leverage the idea of the safety value function. Instead of penalizing the unsafe policy in the objective, we combine the safety value function with a model-based exploration method ATACOM.

## 2    Preliminaries

We formulate the safety problem in the framework of CMDP. A CMDP is defined as a tuple $(\mathcal{S}, \mathcal{A}, \mathcal{P}, r, k, \gamma)$ with a state space $\mathcal{S}$, an action space $\mathcal{A}$, a stochastic function $\mathcal{P} : \mathcal{S} \times \mathcal{A} \times \mathcal{S} \to \mathbb{R}$ that represents the transition probability from a state to another state by an action, a reward function $r(s, a) \in [r_{\min}, r_{\max}]$, a constraint function $k(s) \in [k_{\min}, k_{\max}]$, and a discount factor $\gamma \in [0, 1)$. We first define the safety and feasibility following [28] as:

**Definition 1.** *Consider a constraint function $k : \mathcal{S} \to \mathbb{R}$ and a policy $\pi : \mathcal{S} \to \mathcal{A}$. i. A state $s$ is* Safe *if $k(s) \leqslant 0$. ii). The* Safe Set *is defined as $\mathcal{S}_S = \{s \in \mathcal{S} : k(s) \leqslant 0\}$. iii. The* Unsafe Set *is the complementary set $\bar{\mathcal{S}}_S = \mathcal{S} \backslash \mathcal{S}_S$. iv. A state $s$ is* Feasible *under a policy $\pi$ if $k(s_t) \leqslant 0$ for all $t \in \{0, 1, \ldots, \infty\}, s_0 = s, a_t = \pi(s_t)$.*

The SafeRL problem is formulated as a constrained optimization problem

$$\max_{\pi} \quad \mathbb{E}_{\pi}\left[\sum_{t=0}^{\infty} \gamma^t r(s_t, a_t)\right], \qquad \text{s.t.} \quad \mathcal{F}(s) \leqslant 0, \tag{1}$$

where $\mathcal{F}$, depending on the perspective on the safety problem [32], can take different forms, such as

$$k(s_t) \leqslant 0, \forall t \quad \text{(2a)} \qquad \mathbb{P}(k(s_t) > 0) \leqslant \eta_c, \forall t \quad \text{(2b)} \qquad \mathbb{E}_{\pi}\left[\sum_t \gamma^t k(s_t)|s_t\right] \leqslant \eta_e \quad \text{(2c)}$$

The first constraint in (2a) describes the hard constraint, to be satisfied at each time step. However, we cannot enforce these constraints in the setting of stochastic environments. To address this issue, we can use chance constraints, as shown in (2b), restrict the probability of the violations to be smaller than a threshold $\eta_c$. Both (2a) and (2b) are stepwise constraints that focus on safety at the current time step. The last type of constraint, as shown in (2c), forces the cumulative cost of the trajectory to be smaller than a threshold $\eta_e$. In this paper, we will focus on the long-term safety constraint in a stochastic formulation, a combination of (2b) and (2c).

**Distributional Reinforcement Learning** Unlike the typical RL setting that considers the expected value of the reward (the cumulative cost in our case), distributional RL treats the reward (cost) as a random variable and, therefore, the value function describes the distributions of the random cumulative return (cumulative cost). Distributional RL has shown superior performance in many benchmarking tasks, such as Atari Games [33, 34, 35] and Mujuco tasks [36] since the distributional value function contains more information beyond the first moment. The value function is represented as a random variable $Z^\pi$ instead of a scalar of the expected value $Q^\pi$, the random variable Bellman equation has a similar form [2] $Z^\pi(s) \overset{\mathcal{D}}{=} R(s) + \gamma Z^\pi(S')$ where the distribution of the random variable $S'$ depends on policy $\pi$ and dynamics $\mathcal{P}$. The distribution $Z^\pi(s)$ can be represented by different types of models, such as the network with fixed support [33, 34], Gaussian Network [37] and Implicit Quantile Networks (IQN) [35]. We demonstrate our method using Gaussian Networks. However, our method is not limited to the model type as shown in Appendix C using IQN.

**Safe Learning on the Constraint Manifold** We briefly introduce the ATACOM approach [38, 3, 4], which forms the basis of our approach. ATACOM addresses stepwise hard constraint, as defined in Equation (2a). Furthermore, ATACOM assumes that the dynamic system of the robot is a given nonlinear affine system, $\dot{s} = f(s) + G(s)a$. ATACOM constructs the *Constraint Manifold* by introducing a slack variable $\mu$ as $\mathcal{M} := \{(s, \mu) \in \mathcal{D} : c(s, \mu) = 0\}$ with $c(s, \mu) = k(s) + \mu$. We assume that $\mu$ is equipped with a dynamic system $\dot{\mu} = \alpha(\mu)u_\mu$ and $\alpha$ is a class $\mathcal{K}$ function[3]. Using the concept of Constraint Manifold, a safe controller can be obtained by setting $\frac{d}{dt}c(s, \mu) = 0$. The resulting controller has the following form

$$\begin{bmatrix} a \\ u_\mu \end{bmatrix} = W(s, \mu, a) := -J_u^\dagger \psi - \lambda J_u^\dagger c + B_u u, \quad \text{(3)}$$

with $J_u(s, \mu) = [J_G(s) \quad A(\mu)]$, $J_G(s) = J_k(s)G(s)$, $J_k(s) = \frac{d}{ds}k(s)$ and the Constraint Drift $\psi(s) = J_k(s)f(s)$ induced by the system drift $f(s)$. $A(\mu) = \text{diag}(\alpha_i(\mu_i)), i \in \{1, \ldots, K\}$ is a $K$-dimensional diagonal matrix for the slack variable. $B_u$ is a set of basis vectors tangent to the manifold. The first and the second terms on the right-hand side compensate the drift $\psi$ and retract the system to the manifold; the last term is the tangential term that drives the system along the constraint manifold. An RL agent only needs to learn a policy for the tangential action $u \sim \pi(s)$, while the safety is guaranteed by the controller structure. Alg. 3 in Appendix A showed a detailed process of action mapping. In this work, we extend ATACOM for the chance constraint and propose a new method to simultaneously learn the policy and the long-term constraint online.

## 3 Long-term Safety under Uncertainty

In this section, we will introduce a method to estimate the constraint for long-term safety, and then we show how to integrate the time-varying constraint into the ATACOM learning framework. The overall algorithm Alg. 1 can be found in Appendix A.

---

[2] $A \overset{\mathcal{D}}{=} B$ denotes that two random variable $A$ and $B$ are equal in distributions.
[3] class $\mathcal{K}$ function: (1) continuous; (2) strictly increasing; (3) $\alpha(0) = 0$.

## 3.1 Feasibility Value Function for Long-Term Safety

To ensure long-term constraint satisfaction, we introduce the concept of *Feasibility Value Function (FVF)*, which describes the expected cumulative constraint violation under a policy $\pi$ with an infinity horizon. Formally, we define the feasibility value function under a policy $\pi$ as

$$V_{\mathrm{F}}^\pi(s) = \mathbb{E}_\pi\left[\sum_{t=0}^\infty \gamma^t \max(k(s_t), 0)\,\middle|\, s_0 = s\right] \tag{4}$$

We assume the constraint $k(s) \in [k_{\min}, k_{\max}]$ is bounded and consequently $V_{\mathrm{F}}^\pi \in [0, \frac{k_{\max}}{1-\gamma}]$. When $k$ is an indicator function of the constraint violation, the FVF is analogous to the Constraint Decay Function (CoDF) introduced in [28], defined as $F^\pi(s) = \gamma^{N_\pi(s)}$, with $N_\pi(s)$ the number of steps to the first constraint violation. Unlike the CoDF, which assumes the unsafe state to be absorbing, the FVF does not assume episode termination at the unsafe state, allowing the agent to retract back to a safe state in the future. Therefore, $V_{\mathrm{F}}^\pi(s) \geqslant F^\pi(s)$. The feasibility value function estimates the expected discounted cumulative cost of $\max(k(s), 0)$ under policy $\pi$. We can use the standard Bellman operator $(\mathcal{B}^\pi V_{\mathrm{F}})(s) = \max(k(s), 0) + \gamma V_{\mathrm{F}}(s)$ to update the estimate. For continuous state-action space, a common choice is to use a neural network to approximate the value function and update the value function using TD learning. When $V_{\mathrm{F}}^\pi(s) = 0$, we have $\max(k(s), 0) = 0$ indicating $k(s) \leqslant 0$. Thus, the *Feasibile Set* $\mathcal{S}_F = \{s \in \mathcal{S} : V_{\mathrm{F}}^\pi(s) = 0\}$ is a subset of the Safe Set. Ensuring the feasibility value function to be zero is sufficient to guarantee stepwise safety.

## 3.2 Distributional Feasibility Value Iteration

The original FVF definition in Eq. (4) evaluates the expected value over the future cost. This estimation does not capture the distribution of the future cost and could fail to ensure safety when the distribution is multi-modal or heavy-tailed. Instead, we can exploit the theory of *Distributional RL* [39] that learns the parametric model to approximate the distribution of the future expected cost. Then, we can construct Value-at-Risk (VaR)/Conditional Value-at-Risk (CVaR) constraints that consider both the safety and the uncertainty of the prediction when drawing actions.

Different parametric models have been used to approximate the distribution of the random value function. In this paper, we approximate the target distribution $\max(k(s), 0) + \gamma V_{\mathrm{F}}^\pi(s)$ with Gaussian support up to the 2nd-order moment [37, 31], i.e. we assume $V_{\mathrm{F}}^\pi(s) \sim \mathcal{N}(\mu^F(s), \Sigma^F(s))$. We can compute the mean of the target distribution $\mu^F(s) = k'(s) + \gamma \mu^F(s')$ and the variance

$$\Sigma^F(s) = k'(s)^2 + 2\gamma k'(s) \mathop{\mathbb{E}}_{s' \sim \mathcal{P}^\pi}\left[\Sigma^F(s')\right] + \gamma^2 \mathop{\mathbb{E}}_{s' \sim \mathcal{P}^\pi}\left[\Sigma^F(s') + \left(\mu^F(s')\right)^2\right] - \left(\mu^F(s)\right)^2$$

Here, $k'(s) = \max(k(s), 0)$ and $\mathcal{P}^\pi(s'|s)$ is the transition probability under policy $\pi$. The distribution of FVF is parameterized by Gausian $\mathcal{N}(\mu_\phi^F(s), \Sigma_\phi^F(s))$. The TD error between the target distribution $\mathcal{N}(\mu^F(s), \Sigma^F(s))$ and the parameterized distribution $\mathcal{N}(\mu_\phi^F(s), \Sigma_\phi^F(s))$ with respect to the 2-Wasserstein distance can be computed as

$$\mathcal{L}_F = \|\mu^F(s) - \mu_\phi^F(s)\|^2 + \mathrm{Tr}\left(\Sigma^F(s) + \Sigma_\phi^F(s) - 2\left(\Sigma^F(s)^{1/2}\Sigma_\phi^F(s)\Sigma^F(s)^{1/2}\right)^{1/2}\right). \tag{5}$$

Since the FVF is one-dimensional, Eq. (5) becomes $\mathcal{L}_F = \|\mu^F(s) - \mu_\phi^F(s)\|^2 + \|\sigma^F(s) - \sigma_\phi^F(s)\|^2$. We use a *Softplus* activation for the mean and a *Exponetial* parameterization for the standard deviation to ensure both values are positive. We use the Gaussian parameterization for illustration purposes. However, our method is not restricted to such parameterization. Experiments using IQN [35, 21] can be found in Appendix C.

During training, we keep a replay buffer of limited size. As training progresses, the agent will behave safer and encounter fewer constraint violations, flushing away unsafe transitions when using a single replay buffer. Instead, we would like the agent to remember the failures and avoid being overly optimistic, using a separate, smaller, *Failure Buffer* $\mathcal{D}_f$ to store the unsafe transitions. In each data batch, we sample a proportional number of data coming from this buffer.

### 3.3 Uncertainty-Aware Constraint using (Conditional) Value-at-Risk

VaR and CVaR quantify the risk of a random variable. VaR is the smallest value where the probability of $Z$ is bigger than a $\alpha$ and CVaR measures the mean of the $\alpha$-tail of the distribution.

$$\mathrm{VaR}_\alpha(Z) = \inf\{z \in \mathbb{R} | F(z) \geqslant \alpha\}, \quad \text{(6a)} \qquad \mathrm{CVaR}_\alpha(Z) = \mathbb{E}\left[z | z \geqslant \mathrm{VaR}_\alpha(Z)\right]. \quad \text{(6b)}$$

where $F(z)$ is the Cumulative Distribution Function (CDF) of the random variable $Z$. When $F$ is continuous and strictly increasing, the VaR is uniquely defined as $\mathrm{VaR}_\alpha(Z) = F^{-1}(\alpha)$, i.e., the quantile function. The VaR and CVaR offer a risk-aware constraint formulation by restricting their values to be smaller than a threshold $\delta$. The CVaR constraint for the Gaussian distribution [40] of FVF is

$$\mathrm{CVaR}_\alpha^F(s) := \mu^F(s) + \frac{1}{1-\alpha}\varphi(\Phi^{-1}(\alpha))\Sigma^F(s) \leqslant \delta, \quad \text{(7)}$$

where $\varphi$ and $\Phi$ are the Probability Density Function (PDF) and the CDF of the standard normal distribution, respectively. $\alpha$ determines the probability of constraint satisfaction; thus, the risk is $1 - \alpha$.

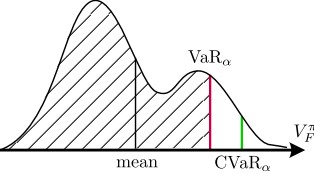

Figure 1: Distribution of $V_F^\pi$ and illustration of mean, VaR (red), and CVaR (green). The shaded area shows the cumulative probability $\alpha$.

Using the CVaR constraints (7), the constraint in problem (1) is defined as $\mathcal{F}(s) := \mathrm{CVaR}_\alpha^F(s) - \delta \leqslant 0$. During exploration, the agent draws an action $u \sim \pi(u|s)$, which is then converted to a safe action $a = W(s, u)$ using ATACOM. Detailed algorithm of ATACOM is shown in Alg. 3 in Appendix A.

**Adaptive constraint threshold estimate** While ideally, we would like to have the FVF to be always equal to zero, setting $\delta = 0$ is neither a practical choice since the network's mean is always bigger than 0, nor beneficial for the training of FVF as it restricts the exploration. The threshold $\delta$ trades off the constraint violation and the exploration and requires further engineering. The experiment comparing different thresholds is illustrated in Appendix E.1.

To alleviate the engineering effort, we propose an adaptive scheme that updates the $\delta$ based on the current episodic cost and the estimation of the FVF. We use a Softplus parametrization to keep the $\delta$ positive. The $\delta$ parameter is updated during the learning process after each episode of horizon $H$ using the following loss

$$\mathcal{L}_\delta = \frac{1}{H}\sum_{i=0}^{H} L_{\text{Huber}}\left(d_c(s_i), \mathrm{CVaR}_\alpha(s_t) - \delta\right) \quad \text{(8)}$$

where the term $d_c(s_i) = \sum_{t=i}^{H} \gamma^{t-i}k'(s_t) - \bar{C}$ computes the difference between the empirical discounted cost starting from $s_i$ and the accepted discounted cost budget $\bar{C}$ and $\mathrm{CVaR}_\alpha(s_t) - \delta$ computes the distance of the estimated FVF to its threshold. We use *Huber Loss* [41] to obtain a more robust update against outliers. As shown in Figure 2, $\delta$ is tuned based on the empirical cost to its cost budget. If $d_c(s_i)$ is bigger than 0 (the actual cost is bigger than the budget), we reduce $\delta$ for a more conservative policy. Conversely, if $d_c(s_i)$ is smaller than 0, we increase $\delta$ to loose the constraints.

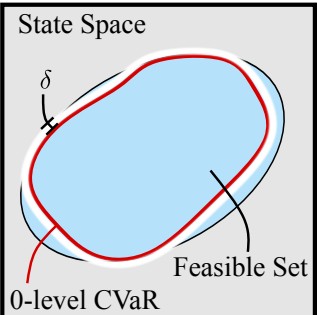

Figure 2: Illustration of the feasible set (light blue), the learned FVF at 0-level *red* and threshold $\delta$. The threshold $\delta$ provides a small feasible region (white) to explore within a small cost budget.

### 3.4 Policy Iteration with Learnable Constraint using ATACOM

As introduced in Section 2, ATACOM constructs a safe action space by determining the basis vectors of the tangent space of the constraint manifold. However, previous work assumes that the constraint is given, fixed, and deterministic. This assumption is no longer valid when the constraint is trained during the learning process. Since the constraint function changes during training, the safe action space changes accordingly, leading to a non-stationary Markov Decision Process (MDP), which

causes the failure of the training. Specifically, let $a \in \mathcal{A}$ be the action applied to the environment and the $u \in \mathcal{U}$ be the control input obtained from the policy $u \sim \pi(s)$. ATACOM constructs an affine mapping $W : \mathcal{U} \rightarrow \mathcal{A}$, defined in (3), that maps an action in safe space into the original one. Combining ATACOM with the actor-critic framework, a value function estimator $Q_\omega(s, u)$ is trained to approximate the expected return. Then, the policy $\pi_\theta(s)$ is updated by maximizing the $Q_\omega(s, u)$. However, when the constraint is updated during the training process, the action mapping $W$ will also change. The same action $u$ will result in different $a$ at different steps. This variation of the action space leads to an unstable update of $Q_\omega(s, u)$ and $\pi_\theta(s)$.

In the following, we will demonstrate how to address this problem for the Soft Actor Critic (SAC) algorithm [42]. However, it is possible to use the same methodology to extend most Deep RL algorithms, e.g., DDPG [43] TD3 [44], PPO [45]. To solve this issue, we learn the value function of the original action space $Q_\omega(s, a)$, which is invariant to the constraints. Since $Q_\omega$ is represented by a neural network and can be differentiated, we can use the reparameterization trick to obtain the gradient for $\theta$ (similar to TD3 and SAC [42]), the objective and policy gradient can be obtained as

$$\max_{\pi_\theta} J^\pi = \mathbb{E}_{s, u \sim \pi_\theta(s)} [Q_\omega(s, W(u))], \qquad \nabla_\theta J_\pi = \nabla_a Q_\omega(s, a) \nabla_u W(u) \nabla_\theta \pi_\theta(s).$$

Note that in SAC, the soft Q-function includes the entropy term $H(W(\pi_\theta(s)))$ to encourage exploration. The entropy term is indeed constraint-dependent. Thus, updating the constraints may change the entropy of the policy, consequently, the estimated value function is not anymore proper. We argue that practically, the variation of the entropy bonus has a negligible effect on the training stability because the entropy term is scaled down by a coefficient. Furthermore, online training in the value function allows us to quickly adapt to the new policy entropy. The overall algorithm of D-ATACOM and the modified SAC algorithm are illustrated in Alg. 1 and Alg. 2 in Appendix A.

## 4 Experiments

In this section, we compare the performance of our approach in three different environments with different characteristics. Details of the environment description can be found in Appendix B. We compare with SafeRL baselines such as the *LagSAC* [46] and the *WCSAC* [31]. All environments and algorithms based on SAC are implemented using the MushroomRL framework [47]. We use the implementations provided by OmniSafe [48] for PPO-based algorithms. We conducted a hyperparameter search on the learning rates, cost budget, and accepted risk with 10 random seeds. We present the results with 25 seeds whose hyperparameters perform the best tradeoff between performance and safety. Further hyperparameter search experiments are in Appendix D.

**Cartpole**  This environment extends the classical Cartpole benchmark. The pendulum is initialized in an upright position, and the goal is to move the pendulum tip to a desired point while keeping the pendulum upright. The constraint enforces an angle smaller than 90 degrees from the upright position. We further added a position limit to the cart. Despite the simplicity of the environment, designing a feasible long-term constraint is very challenging due to the actuator limitation and cart position limits. As we can see from the results in Figure 3, our approach is the best-performing one among the SafeRL baselines in terms of learning speed, while achieving small constraint violations. SAC achieves a policy with higher performance, but this policy heavily violates the safety constraints as it completely disregards the pole angle constraint while reaching the goal. The RCPO algorithm [11] achieves better performance, but violations are comparable with SAC. Notice that D-ATACOM requires a feasible cost budget to generate feasible actions since a single constraint violation will result in a high sum cost, reaching the maximum violation. An unachievable low budget will lead to conservative performance due to a lack of exploration, as shown in Appendix E.2.

**Navigation**  In this task, the goal is to control a differential-driven TIAGo++ robot and learn a navigation policy leading to the goal position while avoiding collision with a moving Fetch Robot. The Fetch robot moves to its randomly assigned target with a moving arm. In this task, agents do not observe the Fetch robot's joint positions while the end-effector's position is available. The safety

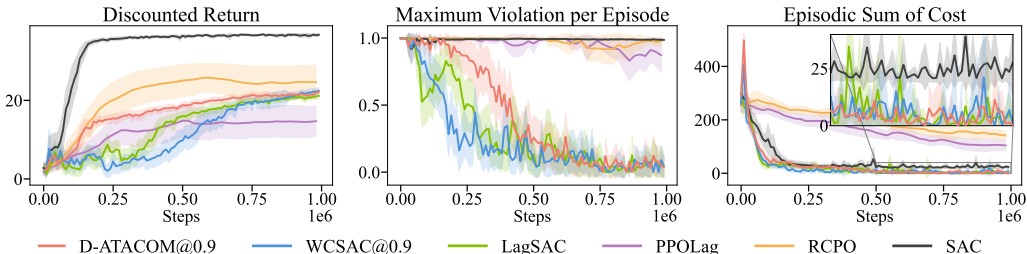

Figure 3: Learning Curves for the Cartpole Environment

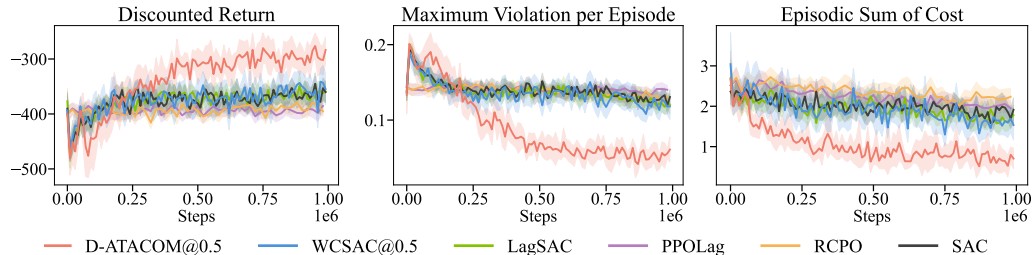

Figure 4: Learning Curves for the Navigation Environment

constraint considers the smallest distance between the two robots. In this setting, the impact of the arm's motion on the constraint is not explicit, which generates a stochasticity on the constraint. From the result in Figure 4, D-ATACOM clearly outperforms all the state-of-the-art methods both in terms of safety and final task performance. Inspecting the learned policies, D-ATACOM is the only approach that performs active collision avoidance behaviors. These collision avoidance behaviors are mostly achieved by the model-based treatment of the constraint function. In this setting, using the constraint gradient and the dynamic model is a strong inductive bias for the method. On top of that, the control system can exploit the physical meaning of the variables, such as other obstacle velocity, allowing it to compensate in advance for the other robot movements. In this task, most approaches behave the same. We argue that the conflict between the task objective and the constraint function, forcing the robot to cause a detour, is problematic for the Lagrangian approaches. Indeed, Lagrangian optimization is trying to balance constraint satisfaction and policy improvement in the update step, possibly causing the algorithm to get stuck in local minima.

**3dof Robot Air Hockey** The objective of this task is to control a 3-DoF arm to score a goal in the robot air hockey task. The arm is controlled by providing acceleration setpoints, and the constraints include the joint position/velocity limits and collision avoidance with the table. Since the optimal strategy for hitting the puck toward the goal is achievable within the tables's boundary, high reward performance and low constraint violations are achievable at the same time.

In addition to common baselines, we also compare D-ATACOM with the original ATACOM with a pre-defined forward invariant constraint (ATACOM+FI) and a non-forward invariant constraint (ATACOM + nFI). As illustrated in Figure 5, D-ATACOM approaches the performance of the AT-ACOM + FI while maintaining low constraint violations. ATACOM + nonFI performs similarly to unconstrained algorithms SAC, both in return and cost, as a poorly defined constraint can not ensure safety. In this task, D-ATACOM is safer than the other baselines at the cost of slower learning performance. The reason for this performance drop is that our approach learns to expand the safe region progressively and improve the performance. The final performance is lower as the robot hits more cautiously at the boundary regions to ensure safety, while other approaches allow the robot to go outside for stronger hitting, as shown in Appendix E.4. A characteristic of this environment is that the constraints do not majorly affect an optimal policy. Therefore, constraint satisfaction is more difficult in the initial phases of learning than in the final one. This allows classical lagrangian methods to be particularly competitive in the task. Figure 6 illustrates the learned constraint at different training steps. We can clearly observe that the feasible region expands progressively and reaches a good coverage at the final epoch. Since FVF is a policy-dependent value function, the prediction at the corner regions is poor, as the policy does not achieve higher performance.

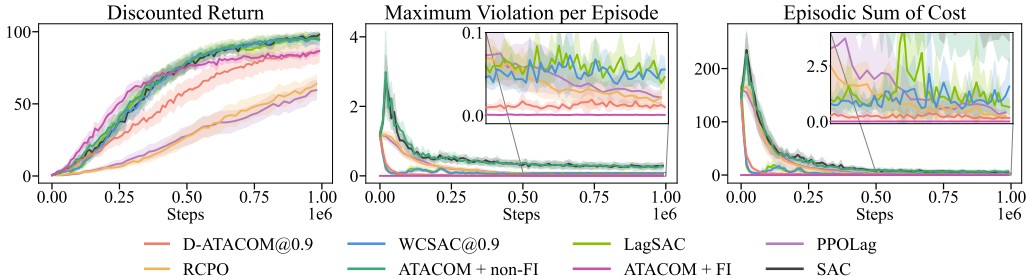

Figure 5: Learning Curves for the Air Hockey Environment

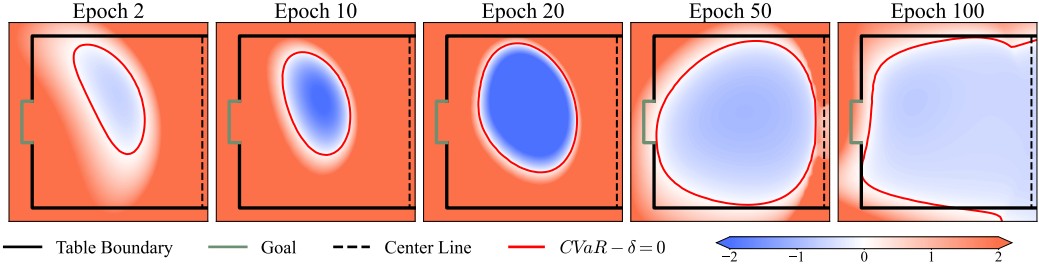

Figure 6: Learned FVF in the Air Hockey task. The color demonstrates the value of the learned constraints $k(s) = \text{CVaR}_\alpha^F(s) - \delta$ in the half of the air hockey table. After a short training with an initial dataset, the feasible region shrinks to a small region at Epoch 2 and then increases progressively, reaching considerable coverage at the end of training.

**Limitations**    While this paper is a first step towards safer and more efficient learning without predefined constraints, the above-mentioned methodologies cannot solve complex control tasks, such as the 7-DoF Robot Air Hockey task with equality constraints. Furthermore, training directly on real robots remains challenging with the current approach, as robots need to explore unsafe states to obtain the FVF. Another limitation of this approach is that it requires knowledge of the robot dynamics. However, even in the presence of model mismatch, the FVF can still impose robust and safe behaviors, as the unsafe transitions caused by the incorrect nominal model are included during training. We show a preliminary study on the robustness of D-ATACOM in Appendix E.5. For this reason, fine-grained domain randomization is a crucial step in successful sim-to-real transfer. Finally, this paper does not explore the possibility of combining constraint learning with known constraints. This can be easily implemented in the D-ATACOM framework. However, the resolution of conflicts between constraints remains to be explored.

# 5   Conclusion

In this paper, we started to bridge the gap between SafeExp methods and model-free SafeRL approaches. We extended the ATACOM framework to work with learned constraints, ensuring long-term safety and properly dealing with constraint uncertainty. Our results show that our method is competitive with state-of-the-art approaches, outperforming them in terms of safety, keeping an on-par learning speed, and achieving similar or better performance for environments with different characteristics. Also, the method does not require excessive parameter tuning, as it includes automatic tuning rules for the most important hyperparameters. Therefore, our work proves that including prior knowledge in data-driven methods can actually be beneficial for scaling SafeRL approaches. Although all of the experiments are trained from scratch, we believe starting with an offline dataset and pre-training will significantly reduce the initial violations, leading to safe performance. In future work, we will further investigate how to integrate known local constraints with long-term safety. This integration will allow scaling the ATACOM approach to real-world robotics tasks involving complex long-term constraints and human-robot interaction.

## Acknowledgments

Research presented in this paper has been supported by the China Scholarship Council (No. 201908080039) and partially supported by the German Federal Ministry of Education and Research (BMBF) within the subproject "Modeling and exploration of the operational area, design of the AI assistance as well as legal aspects of the use of technology" of the collaborative KIARA project (grant no. 13N16274).

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

# A Algorithms

Algorithm 1 demonstrates the D-ATACOM, and Algorithm 2 shows the adapted SAC update used in line 9.

---

**Algorithm 1** D-ATACOM with constraint learning

---

**Initialize:** FVF network $\phi$, number of steps $N$, threshold $\delta$, cost budget $\bar{C}$, policy $\pi_\theta$, value function $Q_\omega$
1: **for** $1 \cdots N$ **do**
2:      Construct $\text{CVaR}_\alpha^F(s_t)$ using $\mu_\phi^F(s_t)$, $\Sigma_\phi^F(s_t)$ from Eq (7).
3:      Draw action $u_t \sim \pi_\theta \in \mathcal{U}$ and obtain safe action $\boldsymbol{a}_t \leftarrow W(s_t, u_t)$ using Alg. 3.
4:      Observe $s_{t+1}, r_t, k_t$ from the environment.
5:      Save replay buffer $(s_t, a_t, r_t, k_t, s_{t+1}) \rightarrow \mathcal{D}$ and $(s_t, k_t, s_{t+1}) \rightarrow \mathcal{D}_f$ if $k_t > 0$.
6:      If the episode terminates, update $\delta$ using Eq. (8).
7:      Sample a batch of transitions $(s, a, r, k, s')$ from $\mathcal{D} \cup \mathcal{D}_f$.
8:      Update $\phi \leftarrow \phi - \alpha_\phi \nabla_\phi \mathcal{L}_F$ using Eq. (5),
9:      Update the value function $Q_\omega$ and policy $\pi$ using SAC in Algorithm 2.
10: **end for**

---

**Algorithm 2** SAC implementation for D-ATACOM

---

**Initialize:** policy parameters $\theta$ and value function parameters $\omega$, learning rate $\eta$
**Input:** Batch of transitions $\mathcal{B} = (s, a, r, k, s')$.
1: Draw action $u'$ and obtain safe action $a'$, $B_u' \leftarrow W(u', s')$ using Alg. 3.
2: Compute log probability $\log p'(a'|s') = \log \pi_\theta(u'|s') - \log|B_u'|$ using the change of variable rule.
3: Update $Q_\omega$ with the TD loss $\mathcal{L}_\omega = 1/|\mathcal{B}|(Q_\omega(s,a) - r - \gamma(Q_\omega(s', a') + \alpha \log p'))^2$.
4: Draw action $u$ and obtain safe $a$, $B_u \leftarrow W(u, s)$ using Alg. 3.
5: Compute log probability, $\log p_\theta = \log \pi_\theta(u|s) - \log|B_u|$.
6: Update policy $\theta \leftarrow \theta - \eta/|\mathcal{B}|\nabla_\theta (Q_\omega(s,a) + \alpha \log p_\theta)$,
     where $\nabla_\theta Q_\omega(s, a) = \nabla_a Q_{\boldsymbol{\omega}}(s, a)\nabla_u W(u)\nabla_\theta \pi_{\boldsymbol{\theta}}(s)$.

---

**Algorithm 3** Construct safe action with ATACOM

---

**Input:** state $s$, action $u$
1: Compute slack variable $\mu$ and constraint $c(s, \mu) = k(s) + \mu$.
2: Compute the Jacobian $J_u(s) = \begin{bmatrix} J_k(s)G(s) & A(\mu) \end{bmatrix}$, drift $\psi(s) = J_k(s)f(s)$.
3: Truncate the drift that have a positive impact on safety $\psi(s) = \max(\psi(s), 0)$.
4: Construct the tangent space basis $B_u$ by computing the kernel of $J_u$ using QR/SVD Decomposition.
5: Compute the safe action $a$ using Eq. 3.
**Output:** safe action $a$, affine mapping matrix $B_u$

---

# B  Experiment Environments

In this Section, we provide the full description of the environments used for the experiments. In all environments, the cost value is a continuous variable. A value greater than zero indicates how much the constraints are violated.

## B.1  Cartpole

The cartpole environment, depicted in Figure 7a is a classic control problem with the goal of moving the pole tip to a desired position (green point) by controlling a cart. The pole is one unit in length and is initialized in an upright position on the cart. The cart can move on a rail 10 units long. The cart is initialized on the left side of the rail, and the goal is to move the cart towards the goal position of the pole tip on the right rail's side while keeping the pole upright.

The state space of the environment is $s = [x, \sin\theta, \cos\theta, \dot{x}, \dot{\theta}]^T$ where $x$ is the position of the cart, $\dot{x}$ is the velocity of the cart, $\theta$ is the angle of the pole with the vertical axis, and $\dot{\theta}$ is the angular velocity of the pole. The action space is $a \in [-1, 1]$ where the action is the force applied to the cart.

The reward function given a goal position $x_G$ and pole tip position $x_T$ is defined as $r(s) = \text{clip}(1 - \frac{\|x_G - x_T\|}{4}, 0, 1)$. The constraint function prevents the pole from deviating more than $\pi$ from the vertical axis. Thus we define the cost function as $c(s) = \max\left(\frac{\theta}{0.5\pi} - 1, 0\right)$.

## B.2  Navigation

The Navigation task consists of two robots, one differential-driven TIAGo++ (white) that moves in a room while avoiding the Fetch robot (blue), as shown in Figure 7b. The Fetch robot constantly moves its robotic arm in a periodic motion, such that the end-effector draws a lemniscate into the air in front of the robot. Additionally, the Fetch robot constantly moves to a randomly assigned target position using a hand-crafted policy that ignores the TIAGo. The agent controls the TIAGo robot to reach the target position while avoiding the Fetch robot, which serves as a dynamic obstacle.

The state space consists of the cartesian position and velocity of the two robots, the target position of the TIAGo, the previous action, and the cartesian position and velocity of Fetch's end-effector. The action space is the linear velocity in the x-direction and angular velocity around the z-axis of the TIAGo robot. These are converted into the left and right wheel velocities.

Given the distance to the goal $d_G$, the current orientation $\theta$ and the goal orientation $\theta_G$ the reward is defined as:

$$r(s) = -\|d_G\| - \text{sigmoid}\left(30(\|d_G\| - 0.2)\right)\frac{\theta_G - \theta}{\pi} - 0.1\|a\|$$

The constraint is the smallest 2d cartesian distance between the TIAGo base and every joint of the Fetch Robot. Additionally, the constraint also prevents the TIAGo from hitting the surrounding walls. Given the TIAGos' position $p_T$ and cartesian position of the ith Fetch joint $p_F^i$ the Fetch cost is $c_F(s) = \max_i(-(\|p_T - p_F^i\| - \omega))$ where $\omega$ is a constant that accounts for the width of the robots. The wall cost is defined as $c_W(s) = \max_i(-(d_{\text{wall}}^i - \omega))$ where $d_{\text{wall}}^i$ is the distance to the ith wall. The step cost is $c(s) = \max(c_F(s), c_W(s))$.

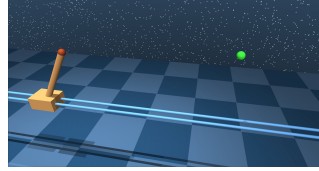
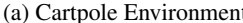
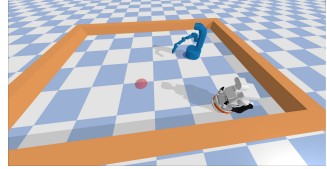
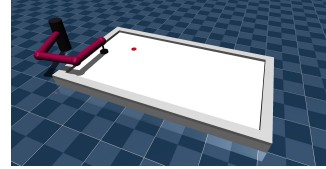

(a) Cartpole Environment          (b) Navigation Environment          (c) Air Hockey Environment

Figure 7: The three Environments used for evaluation of all algorithms

## B.3 Planar Air Hockey

In the Planar Air Hockey environment, the agent controls a 3-DoF robot arm with a mallet attached to the end-effector. The goal is to hit a puck into the opponent's goal, located on the opposite side of the table, as shown in Figure 7c. The episode terminates when the puck enters the goal or hits one of the table's walls.

The state space consists of the robots' joint positions, velocities, puck position, and velocity. The action space is the acceleration setpoint for each robot joint.

The reward for non-absorbing states is the change of distance between the puck and the goal. In absorbing states the reward depends on the distance of the puck to the goal. Given the puck position $[x^t, y^t]^T$ at timestep $t$ and the distance between puck and goal as $d^t$, we define the reward as:

$$
r(s_t) = \begin{cases}
50(d^{t-1} - d^t) & \text{if not absorbing} \\
\rho(1.5 - 5 \cdot \text{clip}(|y^t|, 0, 0.1)) & \text{if puck in goal} \\
\rho(1 - 2 \cdot \text{clip}(|y^t| - 0.1, 0, 0.35)) & \text{if puck on backboard next to goal} \\
\rho(0.3 - 0.3 \cdot \text{clip}(1.43 - |x^t|, 0, 1)) & \text{if puck on sidebars} \\
0 & \text{otherwise}
\end{cases}
$$

where $\rho$ is a constant that scales the reward. The constraint prevents the mallet from touching the sides of the table and the robot from violating its joint position and joint velocity limits. The mallet cost is defined as $c_M(s) = \max_i(-d_W^i + \omega)$ where $d_W^i$ is the distance to the ith wall and $\omega$ is a constant that accounts for the width of the mallet. Given the joint positions $q_i$ and the joint velocities $\dot{q}_i$ the position cost is $c_P(s) = \max_i(q_i - q_{u,i}, -q_i + q_{l,i})$ and the velocity cost is $c_V(s) = \max(\{\dot{q}_i - \dot{q}_{u,i}, -\dot{q}_i + \dot{q}_{l,i})$. The total cost is $c(s) = \max(c_P(s), c_V(s), c_M(s), 0)$

# C   Implicit Quantile Network

IQN is a parametric model representing the quantile function of the distribution, which takes a quantile value $\tau$ as input and outputs a threshold value $z$ so that the probability of $Z$ being less or equal to $z$ is $\tau$. Let $\eta_\phi^\tau(s)$ be the quantile function at $\tau \in [0,1]$ for the random feasibility value at state $s$. The TD error between two samples $\tau, \tau' \sim U([0,1])$ for the transition $(s, a, s', r, k)$ is

$$d_\phi^{\tau,\tau'} = k'(s) + \gamma \eta^{\tau'}(s') - \eta_\phi^\tau(s)$$

The IQN model can be optimized via the Huber quantile regression loss

$$\mathcal{L}_\tau(d) = |\tau - \mathbb{I}\{d\}| \mathcal{L}_k(d), \quad \text{where } \mathcal{L}_k(d) = \begin{cases} d^2/2k, & |d| < k \\ |d| - k/2, & \text{otherwise} \end{cases} \tag{9}$$

In Figure 8 we compare the Gaussian and IQN approaches for the navigation task. In this experiment, both algorithms use the same hyperparameters. The Gaussian approach slightly outperforms IQN in terms of performance and safety. We theorize that the source of the performance difference is the hyperparameters, which are tuned for the Gaussian assumption. The main difference in the constraint estimation is that the Gaussian approach predicts higher uncertainty leading to higher performance and safety in this environment. To achieve the same similar with IQN, the cost budget or accepted risk has to be decreased. We plan to further investigate the performance of IQN-ATACOM in future work, especially in environments providing only sparse cost feedback.

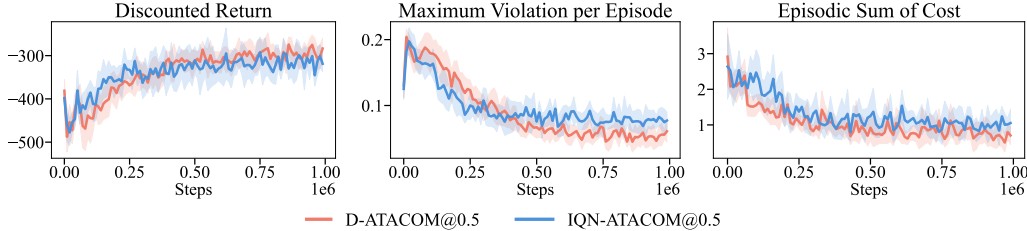

Figure 8: Comparison between the Gaussian distribution assumption and the direct CDF estimation for the navigation task. Both experiments use the same hyperparameters.

# D   Hyperparameter tuning

In this section, we report the parameter tuning for all the baselines in all tasks. In general, we test all the methods with different learning rates, cost budgets and safety parameters to ensure the performance of the baseline is optimal. We report all the hyperparameter configurations we tried and indicate which configuration is used for the main evaluation.

Every algorithm is first evaluated with the learning rates of $1e^{-4}$, $5e^{-4}$ and $1e^{-3}$. To keep the computation reasonable, we use the same learning rate for the actor, the critic, the constraints, and the learning rates for the Lagrangian multiplier that are updated every step. We report the results of these experiments for each task in the following sections.

As a second step, we experimented with different cost budgets to get the best trade-off between safety and performance. Our goal is to get the least constraint violations possible while maintaining reasonable behavior. As we show in Section E.2, setting the cost budget too low can have an impact on the performance with no safety benefit.

Lastly, we tuned the cost-dampening parameters of LagSAC and WCSAC using the same principle we used for the cost budget.

## D.1   CartPole

Figure 9 shows the results of the learning rate tuning for the CartPole task. We can see that RCPO and LagSAC have a learning rate that achieves the best performance. For PPOLag and WCSAC, the differences are more nuanced. Table 1 shows all the parameters we tried for the Cartpole task. The resulting best parameters used for the main evaluation can be found in Table 2.

| | RCPO | PPOLag | LagSAC | WCSAC | D-ATACOM |
|---|---|---|---|---|---|
| **Sweeping parameter** | | | | | |
| learning rate actor/critic/constraint | | | $\{1e^{-3}, 5e^{-4}, 1e^{-4}\}$ | | |
| cost budget | 5 | 5 | | $\{0.1, 5, 25, 40\}$ | |
| cost dampening | - | - | $\{1, 10\}$ | | - |
| learning rate lagrangian multipliers | 0.035 | 0.035 | | $\{1e^{-4}, 5e^{-4}, 1e^{-4}\}$ | |
| accepted risk | - | - | - | | $\{0.1, 0.5, 0.9\}$ |
| **Default parameter** | | | | | |
| epochs | 100 | 100 | 100 | 100 | 100 |
| steps per epoch | 20000 | 20000 | 10000 | 10000 | 10000 |
| steps per fit | 20000 | 20000 | 1 | 1 | 1 |
| episodes per test | - | - | 25 | 25 | 25 |
| network size | | | [128 128] | | |
| batch size | 128 | 64 | 64 | 64 | 64 |
| initial replay size | - | - | 2000 | 2000 | 2000 |
| max replay size | 200000 | 200000 | 200000 | 200000 | 200000 |
| soft update coefficient | - | - | $1e^{-3}$ | $1e^{-3}$ | $1e^{-3}$ |
| warm-up transitions | - | - | 2000 | 2000 | 2000 |
| target kl | 0.01 | 0.02 | - | - | - |
| update iterations | 10 | 40 | - | - | - |

Table 1: Training Parameters for the CartPole task

|  | RCPO | PPOLag | LagSAC | WCSAC | D-ATACOM |
|---|---|---|---|---|---|
| **Sweeping parameter** | | | | | |
| learning rate actor/critic/constraint | $5e^{-4}$ | $1e^{-4}$ | $5e^{-4}$ | $5e^{-4}$ | $5e^{-4}$ |
| cost budget | 5 | 5 | 5 | 5 | 40 |
| cost dampening | - | - | 1 | 1 | - |
| learning rate lagrangian multipliers | 0.035 | 0.035 | $5e^{-4}$ | $5e^{-4}$ | $5e^{-4}$ |
| accepted risk | - | - | - | 0.9 | 0.9 |

Table 2: Result of hyperparameter tuning for the CartPole task

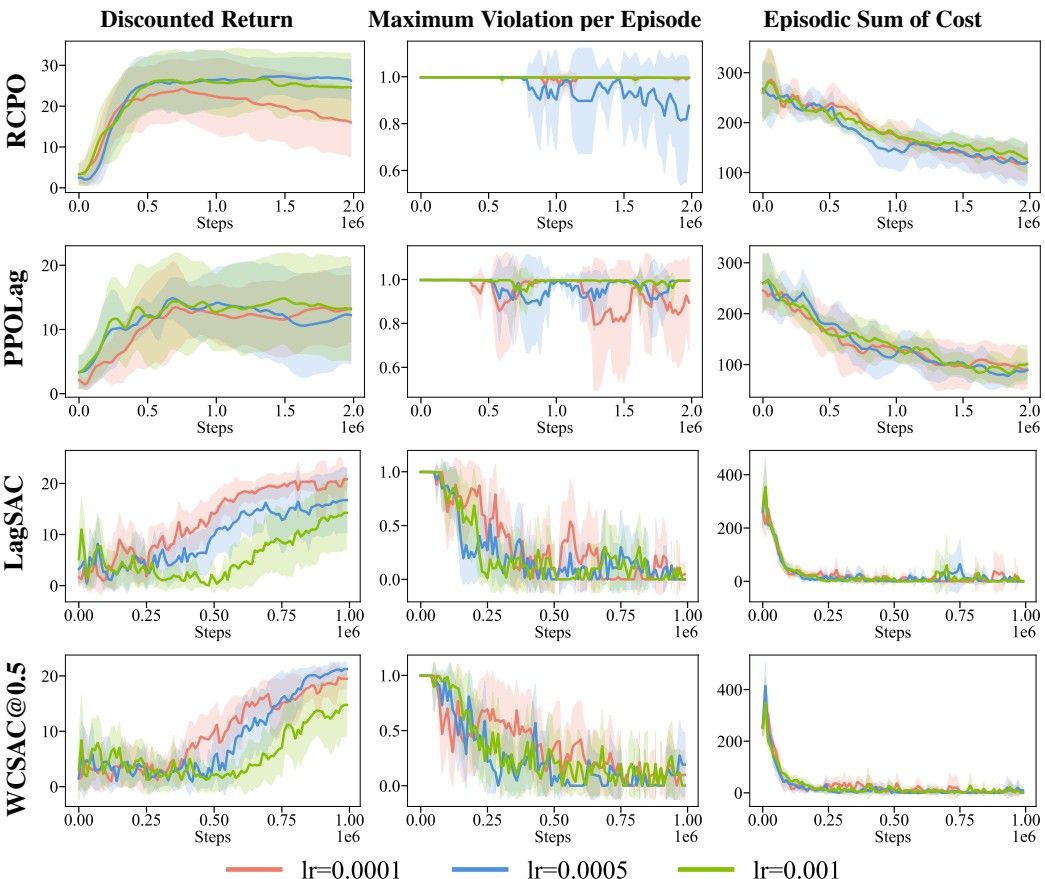

Figure 9: Learning rate ablation study for the Cartpole task. For each experiment we run 10 seeds with all learning rates of the algorithm set to the respective value.

## D.2 Navigation

Figure 10 shows the results of the learning rate tuning for the navigation task. We can see WCSAC is the only algorithm where the learning rate has a significant impact on the performance. Table 3 shows all the parameters we tested for the navigation task. The resulting best parameters used for the main evaluation can be found in Table 4.

| | RCPO | PPOLag | LagSAC | WCSAC | D-ATACOM |
|---|---|---|---|---|---|
| **Sweeping parameter** | | | | | |
| learning rate actor/critic/constraint | | | $\{1e^{-3}, 5e^{-4}, 1e^{-4}\}$ | | |
| cost budget | 0 | 0 | | $\{0, 1\}$ | |
| cost dampening | - | - | | $\{1, 10\}$ | - |
| learning rate lagrangian multipliers | 0.035 | 0.035 | | $\{1e^{-4}, 5e^{-4}, 1e^{-4}\}$ | |
| accepted risk | - | - | - | | $\{0.1, 0.5, 0.9\}$ |
| **Default parameter** | | | | | |
| epochs | 100 | 100 | 100 | 100 | 100 |
| steps per epoch | 20000 | 20000 | 10000 | 10000 | 10000 |
| steps per fit | 20000 | 20000 | 1 | 1 | 1 |
| episodes per test | - | - | 25 | 25 | 25 |
| network size | | | [128 128] | | |
| batch size | 128 | 64 | 64 | 64 | 64 |
| initial replay size | - | - | 2000 | 2000 | 2000 |
| max replay size | 200000 | 200000 | 200000 | 200000 | 200000 |
| soft update coefficient | - | - | $1e^{-3}$ | $1e^{-3}$ | $1e^{-3}$ |
| warm-up transitions | - | - | 2000 | 2000 | 2000 |
| target kl | 0.01 | 0.02 | - | - | - |
| update iterations | 10 | 40 | - | - | - |

Table 3: Training Parameters for the navigation task

| | RCPO | PPOLag | LagSAC | WCSAC | D-ATACOM |
|---|---|---|---|---|---|
| **Sweeping parameter** | | | | | |
| learning rate actor/critic/constraint | $1e^{-4}$ | $1e^{-4}$ | $1e^{-4}$ | $1e^{-4}$ | $1e^{-4}$ |
| cost budget | 0 | 0 | 0 | 0 | 0 |
| cost dampening | - | - | 1 | 1 | - |
| learning rate lagrangian multipliers | 0.035 | 0.035 | $1e^{-4}$ | $1e^{-4}$ | $1e^{-4}$ |
| accepted risk | - | - | - | 0.5 | 0.5 |

Table 4: Result of hyperparameter tuning for the navigation task

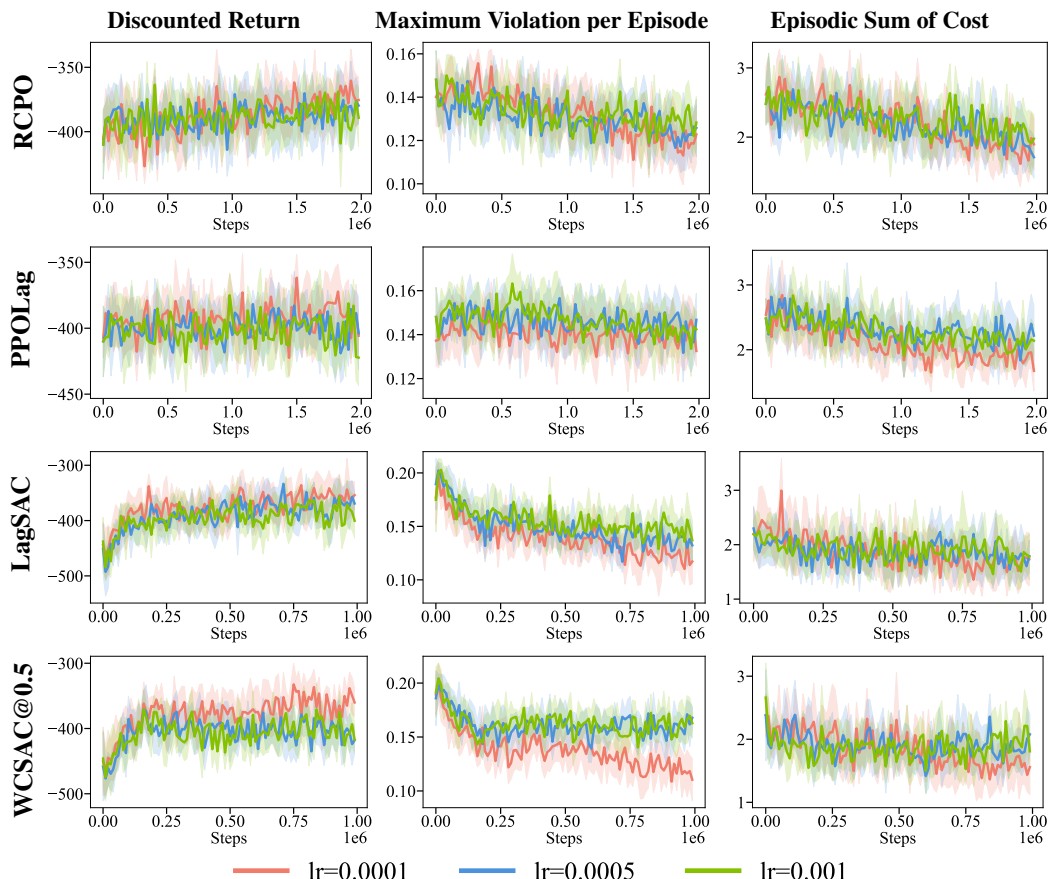

Figure 10: Learning rate ablation study for the Navigation task. For each experiment, we run 10 seeds with all learning rates of the algorithm set to the respective value.

## D.3 Air Hockey

Figure 11 shows the results of the learning rate tuning for the air hockey task. We can see that RCPO and PPOLag learn safer behaviors compared to LagSAC and WCSAC. However, their discounted return is lower, and they need twice as many steps. Table 5 shows all the parameters we tested for the air hockey task. The resulting parameters used for the main evaluation can be found in Table 6.

| | RCPO | PPOLag | LagSAC | WCSAC | D-ATACOM |
|---|---|---|---|---|---|
| **Sweeping parameter** | | | | | |
| learning rate actor/critic/constraint | | | $\{1e^{-3}, 5e^{-4}, 1e^{-4}\}$ | | |
| cost budget | 0 | 0 | | $\{0, 1\}$ | |
| cost dampening | - | - | $\{1, 10\}$ | | - |
| learning rate lagrangian multipliers | 0.035 | 0.035 | | $\{1e^{-4}, 5e^{-4}, 1e^{-4}\}$ | |
| accepted risk | - | - | - | $\{0.1, 0.5, 0.9\}$ | |
| **Default parameter** | | | | | |
| epochs | 100 | 100 | 100 | 100 | 100 |
| steps per epoch | 20000 | 20000 | 10000 | 10000 | 10000 |
| steps per fit | 20000 | 20000 | 1 | 1 | 1 |
| episodes per test | - | - | 25 | 25 | 25 |
| network size | | | [128 128] | | |
| batch size | 128 | 64 | 64 | 64 | 64 |
| initial replay size | - | - | 2000 | 2000 | 2000 |
| max replay size | 200000 | 200000 | 200000 | 200000 | 200000 |
| soft update coefficient | - | - | $1e^{-3}$ | $1e^{-3}$ | $1e^{-3}$ |
| warm-up transitions | - | - | 2000 | 2000 | 2000 |
| target kl | 0.01 | 0.02 | - | - | - |
| update iterations | 10 | 40 | - | - | - |

Table 5: Training Parameters for the air hockey task

| | RCPO | PPOLag | LagSAC | WCSAC | D-ATACOM |
|---|---|---|---|---|---|
| **Sweeping parameter** | | | | | |
| learning rate actor/critic/constraint | $5e^{-4}$ | $1e^{-3}$ | $5e^{-4}$ | $5e^{-4}$ | $5e^{-4}$ |
| cost budget | 0 | 0 | 0 | 0 | 1 |
| cost dampening | - | - | 1 | 1 | - |
| learning rate lagrangian multipliers | 0.035 | 0.035 | $5e^{-4}$ | $5e^{-4}$ | $5e^{-4}$ |
| accepted risk | - | - | - | 0.9 | 0.9 |

Table 6: Result of hyperparameter tuning for the air hockey task

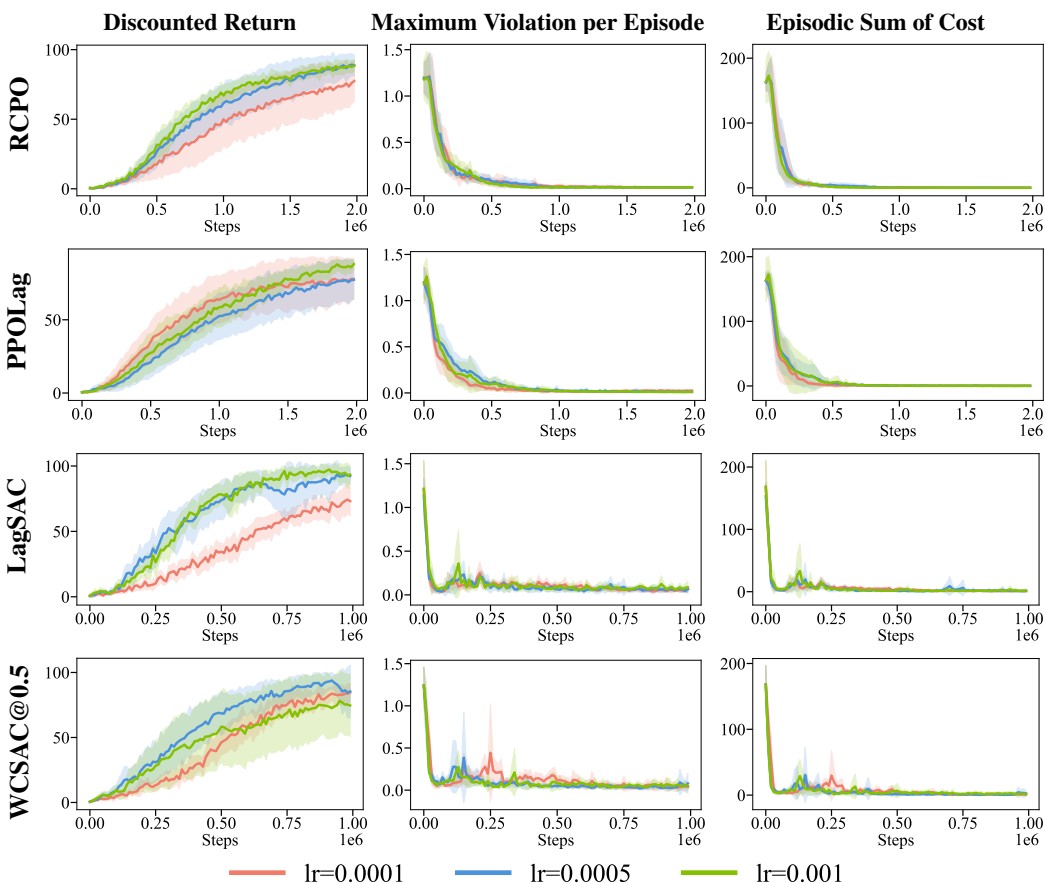

Figure 11: Learning rate ablation study for the Air Hockey task. For each experiment, we run 10 seeds with all learning rates of the algorithm set to the respective value.

# E  Additional Experiments

## E.1  Air-Hockey with Fixed Delta

The safety threshold $\delta$ is an important parameter controlling the trade-off between safety and exploration. A too-small threshold ensures safety at the cost of limiting exploration. Thus, the agent will increase the performance slowly. A higher threshold results in a less restrictive exploration, but the constraint is then not effective to ensure safety.

Therefore, we propose an adaptive threshold that updates its value based on the empirical costs and its prediction of the FVF. In this experiment, we compare D-ATACOM with multiple fixed values for $\delta$ in the planar air hockey task. Figure 12 compares these experiments with the introduced automatic tuning method. We can observe that fixed $\delta$ has a detrimental impact on learning performance because exploration is restrictive by too strict constraints. Interestingly, this lack of exploration also hinders constraint estimation, which leads to slower convergence towards safe behaviors. With a higher fixed $\delta$ there are no exploration issues. However, a higher $\delta$ consequently leads to higher constraint violations. With our adaptive threshold, we get a good trade-off between the learning performance and constraint satisfaction by having an initial high $\delta$ to encourage exploration, which then converges towards a smaller sensible value that reflects a given cost budget.

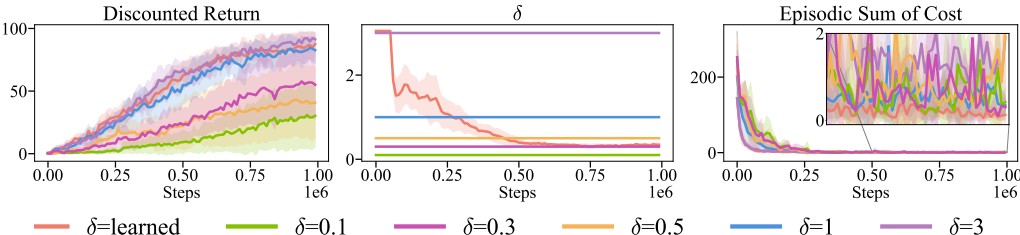

Figure 12: Performance of ATACOM with different fixed $\delta$ values

### E.2 CartPole with different Cost Budget

In this experiment, we will compare the impact of the cost budget parameter on D-ATACOM and WCSAC. We chose the CartPole task for this comparison because both algorithms do not learn a completely safe policy. Figure 13 shows the performance of D-ATACOM and WCSAC with different cost budgets. We can observe that the performance of D-ATACOM is more sensitive to the cost budget parameter compared to WCSAC. When the policy cannot achieve the given cost budget the performance of D-ATACOM degrades significantly. This performance drop occurs because the delta eventually will converge towards zero, which results in a very conservative policy. The behavior for D-ATACOM with the cost budgets of 0.1 and 5 is balancing to the pole in its initial position because the policy is too conservative to move towards the goal, as this will lead to constraint violations.

On the other hand, WCSAC is more robust w.r.t. the cost budget parameter. An unreasonable cost budget will increase the Lagrange multiplier, giving more weight to the constraint. The difference is that the Lagrange multiplier does not set an explicit limit to the constraint like the delta does in D-ATACOM. Instead, WCSAC gives more weight to the constraint violations in the optimization problem, which has less impact on policy performance. Is worth noting that, depending on the application, one of the two behaviors would be preferable. In safety-critical applications, having an algorithm that strongly enforces the constraint violation, independently of the performance, is preferable. Instead, when partial constraint satisfaction is enough, it may be better to choose a lagrangian-based algorithm.

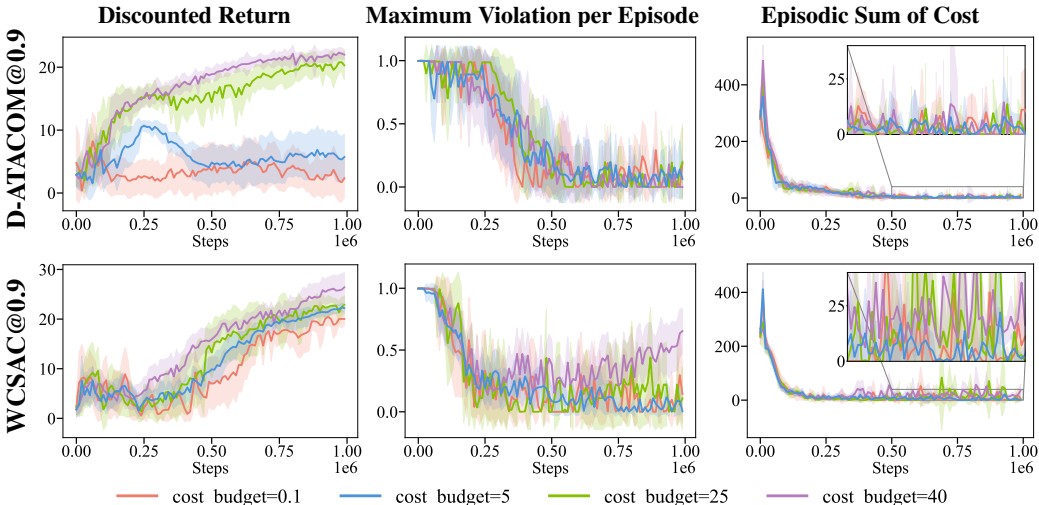

Figure 13: Impact of the cost budget parameter on D-ATACOM and WCSAC performance in the CartPole task

### E.3 Experiment with different Accepted Risk

In the distributional setting, the parameter accepted risk determines how much of the tail of the distribution we are willing to violate, i.e., how much risk we want to take. However, this is not the only parameter that influences the safety of a policy. Usually, there is another parameter that is tuned with a given cost budget that also influences how safe the behavior is. For WCSAC this parameter is the Lagrange multiplier beta, and for D-ATACOM it is the learned $\delta$. To show the complete impact of the accepted risk, we fix $\delta$ to a constant value such that it cannot compensate for the difference in the accepted risk. Figure 14 shows the performance of D-ATACOM with a fixed delta and different levels of accepted risk in the Navigation task. Clearly, a lower accepted risk leads to safer behavior.

The impact of the accepted risk on the safety shrinks for D-ATACOM when the delta is learned. Delta can compensate for a high accepted risk, resulting in the same safe policy as a lower accepted risk would produce. The accepted risk has an impact in this setting toward the beginning of the training when delta is not yet converged. Thus accepted risk determines how risky the exploration at the beginning of the training will be. Figure 15 shows the impact of different accepted risk settings on the air hockey task. Setting a lower accepted risk results in slower exploration, thus achieving a slower convergence of the discounted return. The maximum violation and sum of cost are comparable for all accepted risk settings because, in the air hockey task, the constraint does not majorly affect the optimal policy.

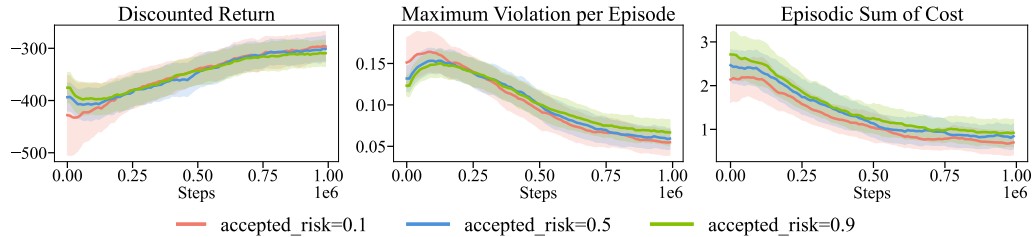

Figure 14: Impact of accepted risk on performance in the Navigation task with a fixed delta. The plots are smoothed via the exponential moving average with 0.9 weight

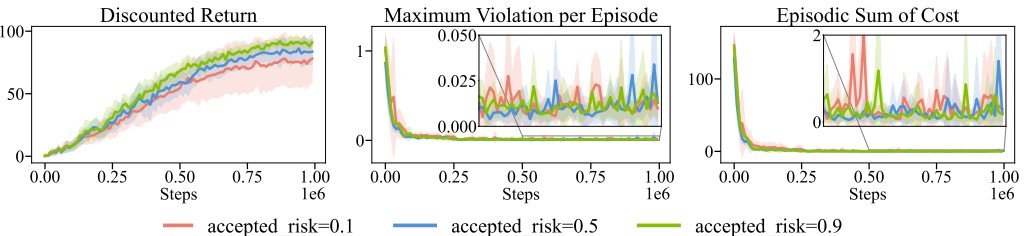

Figure 15: Impact of accepted risk on performance the air hockey task

### E.4 Analysis of Air Hockey

In the air hockey task, D-ATACOM cannot reach the same discounted return as LagSAC and WC-SAC. We investigate the final performance of the policies to understand the differences that lead to the performance gap. As D-ATACOM results in a safer policy, we theorize that performance is lost when the puck is initialized too close to the edge of the table. To test this hypothesis, we evaluate the performance of the final policies with an adjusted region for the initial puck position, that omits these critical positions. Figure 16 shows the performance for the original and adjusted regions and the difference between them.

For the original region D-ATACOM has significant outliers in the discounted return compared to WCSAC and LagSAC. However, LagSAC and WCSAC have more outliers in the maximum violation and sum of cost. Thus, WCSAC and LagSAC sacrifice safety to gain a stable performance. The safe exploration of D-ATACOM results in the opposite behavior, where the policy will sacrifice performance to ensure safety.

When we evaluate the performance with the adjusted region, we can observe that the discounted return of D-ATACOM increases more compared to WCSAC and LagSAC. Additionally, the decrease in maximum violation and sum of cost is more significant for LagSAC and WCSAC. This result confirms our hypothesis that D-ATACOM does not properly hit the puck when it is too close to the edge of the table because it is not possible to do so safely. WCSAC and LagSAC learn to hit the puck in these critical positions, but this comes at the cost of safety.

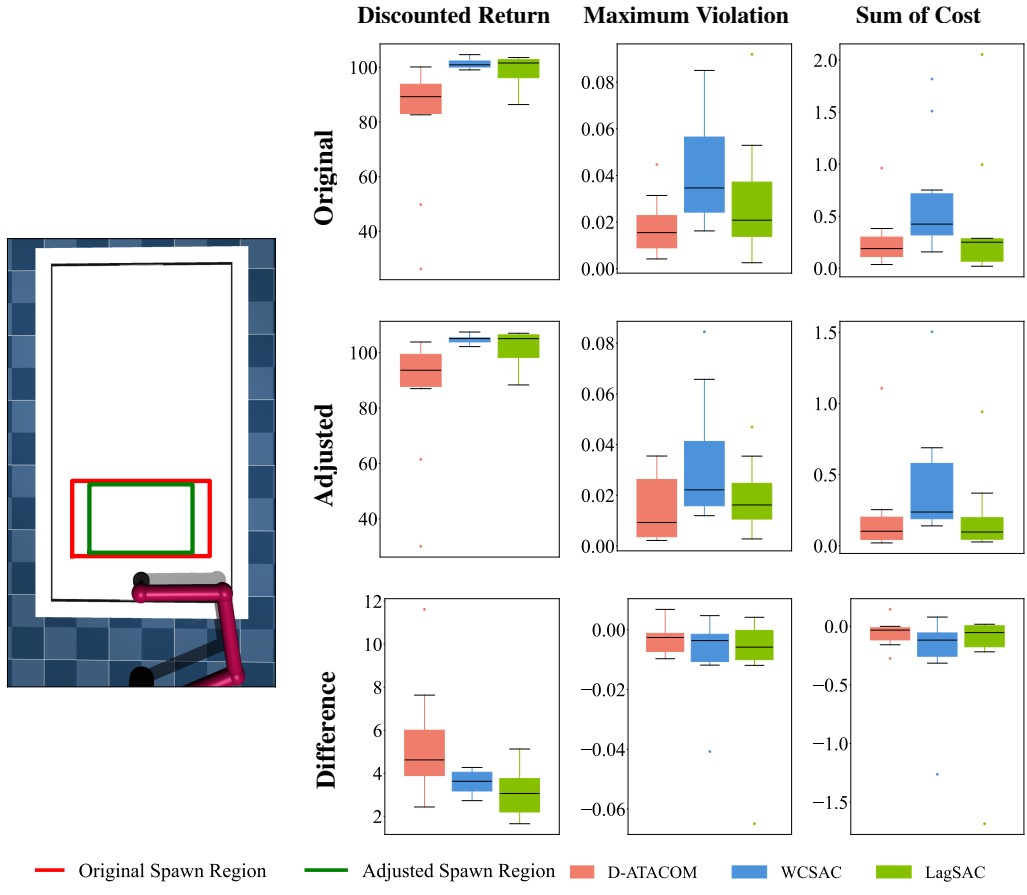

Figure 16: Performance of the final policy from D-ATACOM, WCSAC, and LagSAC in the air hockey task. The performance is evaluated with the original and an adjusted region for the initial puck position.

## E.5 Experiment with Dynamic Model Mismatch and Disturbances

The D-ATACOM exploits the knowledge of the dynamics model to derive the safe policy. The model mismatch could potentially lead to unsafe behaviors. However, D-ATACOM is able to ensure safety under model-mismatch, as the FVF is learned from the dataset collected from the mismatching environment. In this section, we empirically show how dynamic model mismatch will affect the performance of the algorithm.

We use the 3-DoF Air Hockey task as the study example. The robot is controlled by acceleration, we use a simple dynamics model

$$\ddot{\boldsymbol{q}} = \boldsymbol{a}$$

To simulate the model mismatch and disturbances, the dynamics model used in the simulator is

$$\ddot{\boldsymbol{q}} = \boldsymbol{a} - \sigma \dot{\boldsymbol{q}} + \epsilon$$

where $\epsilon \sim \mathcal{N}(0, \sigma)$ is the Gaussian disturbance with standard deviation $\sigma$, and $-\sigma\dot{\boldsymbol{q}}$ is an unmodelled damping term. The experiment comparing the effect of different $\delta$ is shown in Figure 17.

The magnitude of the noise heavily affects the performance of the discounted return, as the RL agent converges to a more robust policy that is more conservative. However, the safety performance remains consistent. This is because the FVF is learned directly from the noisy simulator, and the adaptive threshold balances the exploration and empirical safety during the learning process.

Training directly on real robots remains challenging with the current approach, as robots need to explore unsafe states to estimate the FVF. However, we believe our approach can achieve zero-shot transfer when trained in a realistic simulator. Moreover, since FVF is independent of the dynamic model and the adaptive threshold estimation, we are able to exploit domain randomization during the training process, leading to safe policies that are robust to dynamic mismatches.

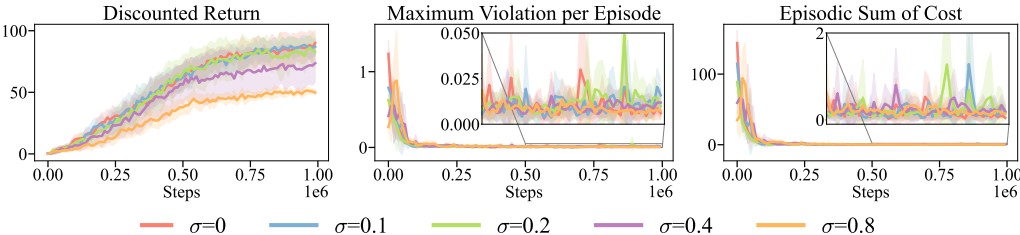

Figure 17: Impact of model mismatch and disturbances of D-ATACOMs in the air hockey task. We add a zero mean Gaussian disturbances with standard deviation $\sigma$ and an unmodelled damping term of the form $-\sigma\dot{q}$ to the environment.

### E.6 Model-Based Baselines

In this experiment, we compared D-ATACOM with other model-based approaches in the Planar AirHockey task. We use the OmniSafe [48] implementation of SafeLOOP [49], which learns the dynamic model and the policy jointly at the same time. Additionally, we implement CBF-SAC, which learns the Control Barrier Function and the policy jointly and tries to ensure the forward invariant property via the agents dynamics. This approach is an adaptation of the work presented in [50] to the online reinforcement learning setting. Finally, we use SafeLayerTD3, which is an exploration method that learns the constraints that encode the dynamics [51].

As shown in Figure 18, CBF-SAC and SafeLayerTD3 achieve on-par performance to D-ATACOM in discounted return. CBF-SAC manages to reduce constraint violations throughout training and converges to a reasonable safety level. However, it converges slower and towards a less safe solution compared to D-ATACOM. SafeLayerTD3 struggles to ensure safety properly because it learns the constraint using supervised learning. This approach fails when the target constraint does not induce long-term safety. SafeLOOP uses a learned dynamics model and look-ahead planning to check safety. Due to the large computational demand, we run the experiment for only 500.000 steps, which is enough for SafeLOOP to converge to a safe behaviour. However, SafeLOOP achieves poor performance in terms of discounted return as it learns a very conservative policy that barely moves the agent. In summary D-ATACOM achieves the same or better performance with stricter adherence to constraints compared to the model-based baselines.

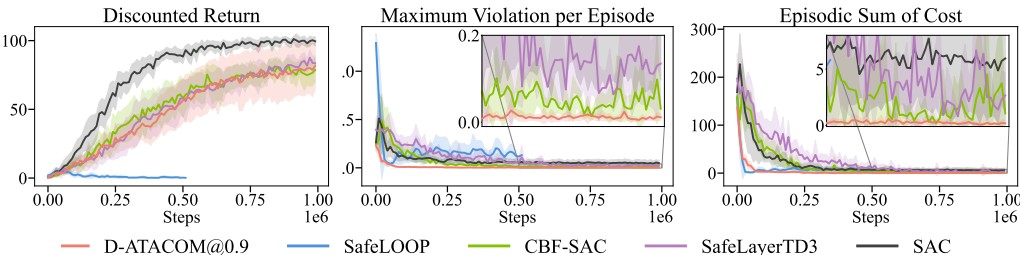

Figure 18: Learning Curves for Model-Based algorithms in the Air Hockey Environment

