# OpenReview forum: "Handling Long-Term Safety and Uncertainty in Safe Reinforcement Learning"
_robot-learning.org/CoRL/2024/Conference — CoRL 2024_

### Official Review · Reviewer_2U5V · 2024-07-22
**Interesting idea and compelling results, could improve on clarity and strength of ablations.**

**Originality:** 3
**Technical Quality:** 3
**Clarity Of Presentation:** 3
**Potential Impact:** 3
**Recommendation:** 3
**Confidence:** 3

**Review:**

Strengths:
- The approach is well motivated, and strikes an interesting balance in the field of Safe RL by using system dynamics knowledge to constrain the policy during exploration to actions whose CVaR on the safety value is estimated to be below a threshold. In this way, the approach achieves lower costs during training than penalty-based baselines. In addition, the approach addresses the realistic setting where agent-dynamics are known but the true safety value function must be learned through interaction (e.g., due to stochasticity in environmental dynamics).
- The results are compelling, showing that D-ATACOM can indeed achieve faster learning than SafeExp methods while having fewer violations per episode compared to Lagrangian penalty based constrained RL methods.

Weaknesses:
- I found the discussion of how the learned feasibility value function is used to define the constraint manifold unclear. Laying out this connection more explicitly in the document would make the paper significantly easier to follow.
- While the approach does strike a balance between dynamics-informed constrained safe exploration and penalty-based constrained RL formulations, the specific balance that is struck seems to be difficult to control. It seems that choosing a fixed $\delta$ risk level is not recommended, as the authors propose the Huber-loss based method to choose $\delta$ as a function of a cost-budget parameter. However, the ablation results varying the cost budget in the appendix seem to suggest that the cost-budget parameter can have a significant impact on discounted return, but does not have a significant impact in the violations per episode or the episodic sum of costs. This suggests that the interpretation of this parameter as defining a budget on the cost does not necessarily hold true in practice? What value does the Huber-loss based adaptive selection of $\delta$ provide over a fixed $\delta$? How does the scale of this loss impact the algorithm's performance?

**Quality Of The Limitations Section:**

3

**Questions For Rebuttal:**

Comments:
- What is $A(\mu)$ defined after equation (3)?
- How is the FVF used to construct the constraint manifold (and ultimately, the action mapping $W$)?
- In Figure 6, what does the color represent? I'm assuming this is $CVaR - \delta$  but this should be stated more clearly.

**Robotics Focus:**

2

**Summary Of Paper:**

This paper presents D-ATACOM, an algorithm combining the SafeExp and CMDP approaches towards Safe RL with the goal of minimizing safety violations during training while still converging towards a constraint-satisfying policy efficiently. To do so, the authors extend the ATACOM for SafeExp by jointly learning the safety value function during training and accounting for uncertainty and stochasticity in the safety constraint when defining the Constrained Manifold.

**Summary Of Recommendation:**

I believe the community would find this paper interesting, as it presents a novel angle to the problem of Safe RL which could be relevant to robotics.

---

### Official Review · Reviewer_PBPX · 2024-07-23
**Important problem and promising approach. The algorithm not clearly written, and the results are not clearly connected to the main title of the paper.**

**Originality:** 3
**Technical Quality:** 2
**Clarity Of Presentation:** 2
**Potential Impact:** 3
**Recommendation:** 3
**Confidence:** 3

**Review:**

The setting of the problem where the model of the robot is assumed to be known and the constraints are assumed to be unknown is realistic, and in this sense, I believe that the authors are tackling an important problem.

The summary of ATACOM is appropriate and clear. Also, Section 3.1 and 3.2 where the feasible value function and its learning scheme using the CVaR concept is overall written well.

However, I could not clearly understand Section 3.3 where the ATACOM is combined with the value function learning. In particular, I think it would be good if the authors include the algorithm table that incorporates every proposed components.

Also, it is not clear to me why the concept of CVaR is necessary for addressing the nonstationary or long-term safety, the main theme of the paper. Relevant to this point, another limitation of the manuscript is that as opposed to the title of the paper, the experiments are not well designed to tackle “long-term” safety. The second experiment does tackle the non-stationary safety constraint, however, in the other two experiments, the safety constraints are just static, which does not differentiate this work much from existing works on learning safety.

Finally, the experiments are compared to model-free methods, whereas the proposed method assumes known dynamics of the robot. Thus, it only makes sense that the proposed method results in better training curves. It will be more informative if the method is compared to existing model-based methods (if there is any).

Minor comments:

- p2 - “we will focus on the constraints in the form of (2c), but in the same stochastic formulation of (2b).”: What this means is not clear.
- Line 153 - reason / reference to why p-Wasserstein distance is used is not clear.
- eq (7) - reference?
- Line 197-200 - It will be better if the description of d_c appears first.
- Line 200 - Huber loss - reference?

**Quality Of The Limitations Section:**

3

**Questions For Rebuttal:**

- Can ATACOM deal with control bounds? Also, if the B_u term in equation (3) vanishes, what happens? (This looks similar to the high relative degree problem of Control Barrier Functions.)
- It is not clear to me why the concept of CVaR is necessary for addressing the nonstationary or long-term safety, the main theme of the paper.
- It would be good to have a pseudo-code of the algorithm in the rebuttal so that I can understand the learning method in more detail.

**Robotics Focus:**

2

**Summary Of Paper:**

The authors address a problem of safety with unknown constraints. To address this, the authors extend the existing framework of ATACOM to a setting where the constraint has to be learned. The ATACOM is combined with learning of the feasibility value function, using the distributional method concepts like VaR and CVaR.

**Summary Of Recommendation:**

The paper needs to improve clarity of the main algorithm, with a clear justification how the proposed scheme solves the problem the paper is addressing. Moreover, the results have to be more convincing that the proposed method can solve nonstationary and long-term safety problems effectively.

---

### Official Review · Reviewer_ydKm · 2024-07-24

**Originality:** 3
**Technical Quality:** 3
**Clarity Of Presentation:** 2
**Potential Impact:** 2
**Recommendation:** 2
**Confidence:** 4

**Review:**

Strength:
 - The paper addresses an important problem: safe reinforcement learning. The setting is particularly challenging, given that the constraint is partially unknown.

Weakness:
 - Although the paper eliminates the assumption of known constraints, which is very important, it requires known dynamics and struggles to scale up to real-world robotics tasks. While assuming some level of knowledge about the dynamics is reasonable, the limited scalability to real robotics applications limits the significance of the contribution. Nevertheless, I appreciate that the authors acknowledged this limitation.
 - The overall algorithm is unclear. A significant portion of the paper is dedicated to problem formulation and preliminaries, but how each part is learned and linked together is not well-explained. Including an overall pseudo-algorithm for clarification would be beneficial.
 - The experiments did not use ATACOM as one of the baselines. Comparing the proposed method with ATACOM is important because ATACOM is the foundational method upon which this paper builds. Including this comparison would help readers better understand the contribution of the proposed approach.

**Quality Of The Limitations Section:**

3

**Questions For Rebuttal:**

- Could you provide a general algorithm to show how the different parts of the method are connected?

 - Is it possible to deploy this method in a real robot setting? If not, what are the potential ways to explore deployment?

 - How does D-ATACOM compare to ATACOM?

**Robotics Focus:**

2

**Summary Of Paper:**

The paper focuses on safe reinforcement learning, building on ATACOM and proposing distributional-ATACOM to integrate dynamics knowledge with constraint learning. Specifically, it addresses long-term safety by estimating a distributional value function for constraint violation. Experiments in Cartpole and navigation demonstrate that the proposed method achieves a lower constraint violation rate, albeit at the cost of a reduced learning rate.

**Summary Of Recommendation:**

The paper addresses an important problem, but the method requires further clarification. Additionally, the discussion on how the method can be deployed in real-world scenarios needs to be expanded.

---

### Author Rebuttal · Authors · 2024-08-07

We have uploaded the revised version of the paper addressing the reviewer's concern. However, some of the experiments are still in progress, and we would like to start the discussion with the reviewer sooner to address further concerns. We will update the paper when all experimental results are ready.

---

### Decision · Program_Chairs · 2024-09-04

**Decision:**

Accept

**Comment:**

**Post-rebuttal metareview**:

This paper extends the ATACOM framework by learning a feasibility value function for solving safe RL problems with known dynamics but unknown constraints. The setting is quite challenging and novel, and the proposed method is convincing. Some reviewers' concerns were addressed in the rebuttal phase. Adding real-world experiments and comparisons with model-based safe RL approaches will strengthen the paper.

------------
**Pre-rebuttal metareview**:

Strengths:
1. Safe RL with unknown constraints is an important and interesting setting.

Weakness:
1. The proposed method requires knowing true dynamics and cannot easily scale up.
2. Need better presentation, especially for algorithms.
3. Need more baselines.
4. The experiment design doesn’t fully show the main point of the proposed method.
5. Other issues mentioned by reviewers.